# Molecular Insight into the Self-Assembly Process of Cellulose Iβ Microfibril

**DOI:** 10.3390/ijms23158505

**Published:** 2022-07-31

**Authors:** Tran Thi Minh Thu, Rodrigo A. Moreira, Stefan A. L. Weber, Adolfo B. Poma

**Affiliations:** 1International Center for Research on Innovative Biobased Materials (ICRI-BioM)—International Research Agenda, Lodz University of Technology, Zeromskiego 116, 90-924 Lodz, Poland; 2Faculty of Materials Science and Technology, University of Science—VNU HCM, 227 Nguyen Van Cu Street, District 5, Ho Chi Minh City 700000, Vietnam; 3Vietnam National University, Ho Chi Minh City 700000, Vietnam; 4BCAM, Basque Center for Applied Mathematics, Mazarredo 14, 48009 Bilbao, Bizkaia, Spain; razevedo@bcmath.org; 5Biosystems and Soft Matter Division, Institute of Fundamental Technological Research, Polish Academy of Sciences, Pawi ńskiego 5B, 02-106 Warsaw, Poland; 6Max Planck Institute for Polymer Research, Ackermannweg 10, 55128 Mainz, Germany; webers@mpip-mainz.mpg.de

**Keywords:** cellulose I, self-assembly, stability, molecular dynamics, Charmm36, β-D-glucose

## Abstract

The self-assembly process of β-D-glucose oligomers on the surface of cellulose Iβ microfibril involves crystallization, and this process is analyzed herein, in terms of the length and flexibility of the oligomer chain, by means of molecular dynamics (MD) simulations. The characterization of this process involves the structural relaxation of the oligomer, the recognition of the cellulose I microfibril, and the formation of several hydrogen bonds (HBs). This process is monitored on the basis of the changes in non-bonded energies and the interaction with hydrophilic and hydrophobic crystal faces. The oligomer length is considered a parameter for capturing insight into the energy landscape and its stability in the bound form with the cellulose I microfibril. We notice that the oligomer–microfibril complexes are more stable by increasing the number of hydrogen bond interactions, which is consistent with a gain in electrostatic energy. Our studies highlight the interaction with hydrophilic crystal planes on the microfibril and the acceptor role of the flexible oligomers in HB formation. In addition, we study by MD simulation the interaction between a protofibril and the cellulose I microfibril in solution. In this case, the main interaction consists of the formation of hydrogen bonds between hydrophilic faces, and those HBs involve donor groups in the protofibril.

## 1. Introduction

One of the most abundant biopolymers on Earth is cellulose, which is the result of the self-assembly process of long linear chains formed by β-D-glucose monomers. The synthesis of these long chains is typical in higher plants [1] and bacteria [2]. In general, their biosynthesis occurs inside the cellulose synthase complex (CESA) [3], where the conversion of a single β-D-glucose monomer to a cellulose polymer requires several enzymatic steps. In bacteria, the mechanism starts with the phosphorylation and formation of uridine diphosphate glucose (UDP-glucose) by UDP-glucose pyrophosphorylase. In the intracellular part of the CESA complex, active UDP-glucose becomes polymerized and gives rise to β-D-glucose chains. The chain is secreted through a transmembrane protein channel. Recently, similar protein complexes were reported in plant cells [1]; similarly, cellulose chains are part of the extracellular product. However, the role of cellulose in plants and bacteria is relatively different and ranges, for instance, from adhesion in bacteria to structural reinforcement and protection in high plants. Furthermore, in nature, the cellulose chains differ in their degree of polymerization and crystal structure. Due to the hydrophilic nature of cellulose chains, they follow aggregation into semi-flexible protofibrils with smaller cross-sections and end by crystallizing into a microfibril with about 36 chains. Cellulose I microfibrils have been observed under X-ray and NMR spectroscopy and found in two different allomorph forms (i.e., Iα and Iβ) [4,5] that are in coexistence. The organization of a few hydrophobic microfibrils into a cellulose bundle can be obtained by water–microfibril interaction, which is a dominant force at large length scales.

The chemical and physical properties of the molecular assembly of cellulose have been the focus of intensive research over the last few decades. Molecular dynamics (MD) simulation has been an invaluable tool in describing the inner structure of solvated polysaccharides and cellulose I microfibrils in water and ionic liquids [6,7,8,9] and in studying large conformational changes, such as the interconversion process between native cellulose I allomorphs and the origin of the cellulose twist in microfibrils [10,11,12], as well as changes between other more stable forms in the laboratory (e.g., I→II and I→IIII) [13], which have a more industrial purpose. Due to the renewable and biodegradable character of cellulose, it has become the main target of several industries, from paper making to biotechnology. They have focused on the chemical modification and efficient extraction of crystalline cellulose I microfibrils from lignocellulose [14]. The energy sector has also devoted major attention to biofuel production using the enzymatic degradation of cellulose microfibrils for posterior conversion into small oligomers for saccharification and final fermentation in bioethanol production [15]. In such cases, the efficiency of large-scale industrial processes has been hampered due to the recalcitrant nature of cellulose microfibrils, which are mostly dominated by molecular networks of hydrogen bonds (HBs) between β-D-glucose chains established by the electrostatic interaction of negatively-charged acceptors and positively-charged donors [16], and also due to the hydrophobic interactions present in crystalline cellulose [16,17,18]. This is an intrinsic feature of the cellulose microfibril that makes its degradation very slow under highly crystalline conditions. In addition, the rapid aggregation of released (longer) oligomers under enzymatic treatment, which can bind to the pretreated cellulose microfibril, is among the limitations that can make enzymatic processing very inefficient [17,18]. In this context, our computational work aims to provide an understanding of the self-assembly process from a molecular point of view that will assist in cellulose biosynthesis and in the pretreatment of biomass, which generally releases different cellulose microfibrils, fibers, bundles, oligomers, etc. In later stages, the microfibril can be used in biofuel production or in materials research for the design of novel cellulose-based materials (e.g., composites) with exceptional mechanical and structural properties [19,20]. In order to control and optimize each of those processes, one has to understand the role played by the molecular fluctuations and hydrogen bond formation that are key to those processes.

Several MD simulations have addressed the structure and stability of the cellulose I microfibril [21], the oligomer decrystallization process [22], and the insolubility of small oligomers on cellulose surfaces [23]. Recently, MD simulation was able to capture the assembly of a cellulose bundle [24], where microfibril–water interactions are mostly responsible for the assembly. In this computational work, we shed light on the physicochemistry of oligomer–microfibril assembly in time scales accessible to MD simulations. We answer questions regarding the molecular preference of the oligomer when binding either to the more hydrophobic microfibril crystalline faces, which are dominated by the presence of the non-polar C–H groups in the (100) and ($\bar{2}$00) crystalline planes, or to the more hydrophilic faces determined by strong interactions with polar hydroxyl (−OH) or hydroxymethyl (–CH_2_OH) groups given by the $\left(\bar{1}10\right)$, (110), $\left(1\bar{1}0\right),$ and $\left(\bar{1}\bar{1}0\right)$ crystal planes (see Figure 1). These two molecular features are responsible for the lateral packing of cellulose chains via stronger interchain HBs (i.e., OH⋯O), which form extended cellulose sheets and then some weak intersheet HBs (i.e., CH⋯O) to assist the stacking of two parallel cellulose sheets in the microfibril. The balance between these two interactions is a crucial step at the early aggregation stage and may impact the efficient assembly process of the crystalline cellulose microfibril. Our study discusses the characterization of the interaction between oligomers and a cellulose I microfibril and analyzes the relevant driving forces during crystallization. Oligomers bound on the microfibril surface undergo conformation changes along the main backbone. After that, interaction with a crystalline surface allows their crystallization. Addressing these phenomena requires the examination of many energetic contributions, such as the van der Waals (vdW) forces and electrostatic contributions for a single chain (oligomer) both in solution and once the chain is adsorbed into the microfibril. Furthermore, we also present a quantification of the most relevant HB types between the oligomer and crystalline microfibril, i.e., those that suggest stability in our MD trajectories and support further crystallization.

We examined the aggregation process of four oligomers and a protofibril (with 18 cellulose chains) with respect to a cellulose I microfibril. The length of the oligomer varied as follows: 6-mers, 12-mers, and 20-mers. The protofibril had a length of 20 β-D-glucose monomers. The aim was to quantify the energetics of the crystallization process for all these systems, including the interplay between oligomer–microfibril complexes as a function of chain flexibility during this process (see Figure 1). Each of these oligomers was relaxed in all-atom MD and simulated from the solvated (disorder) state to the absorbed (semi-ordered) state on the surface of a microfibril of cellulose Iβ. The measure of the crystallization was achieved by the formation of strong HBs, such as O2H⋯O6 and O6H⋯O2, between the oligomer and the microfibril. Once the oligomer reached a crystalized condition, we carried out an energy analysis between the cellulose chain, which was located in the outer layer of the microfibril engaged in the interaction, and the bound oligomer. The oligomers were not restrained during the MD simulation, so they were able to perform a vast search of conformations that enabled the improved stability of the oligomer–microfibril complex.

## 2. Result and Discussion

In order to shed light on the self-assembly process, we focus on the role played by molecular fluctuations in Section 2.1. The information about the change in the oligomer stiffness is the first evidence of absorption and further crystallization on the crystalline surface. In Section 2.2, we also provide a quantification of the energy landscape of the crystallization process. Finally, the strong formation of HBs in Section 2.3 highlights the preferences of oligomer/protofibril when they are in contact with a crystallized microfibril.

### 2.1. Oligomer–Microfibril Interaction: Relevance of the Chain Stiffness during Binding

The characterization of the binding between β-D-glucose oligomers and cellulose I microfibrils through MD simulations revealed a systematic change in chain stiffness for the longer oligomers. Figure 2 shows the RMSF plots for 6-mers. Here, we can see an average RMSF value per monomer below 0.2 nm. The smaller oligomer does bind to the microfibril, with an overall effect that does not affect its internal flexibility. Figure 3 and Figure 4 show the results for longer oligomers, with 12 and 20 β-D-glucose monomers, respectively. The 12-mers case is an intermediate state where the flexibility of the whole chain decreases upon binding, with an RMSF value in the range of 0.3–0.5 nm per monomer. For this case, we can observe that all four oligomers are bound, and monomers in the central part of the chain generally interact strongly with the cellulose fibril, while the ending residues generally present larger fluctuations. The 20-mers case (see Figure 4) shows two oligomers attached to the microfibril surface. For the non-bound cases, we can see large-chain RMSF values in the vicinity of 1.5 nm; for the absorbed oligomers, the RMSF value is quite similar to that of the 12-mers case.

### 2.2. Energetic Characterization of the Binding Process: Electrostatic and vdW Energies

Figure 5 shows the non-bonded energetic contribution to the binding process of the smallest and stiffest oligomer (i.e., six β-D-glucose monomers). In the case of oligomer-1, which is positioned above the hydrophobic crystalline layer (100), we can identify the recognition process as being controlled by the interaction with two cellulose chains that belong to the hydrophilic $\left(1\bar{1}0\right)$ crystal plane, denoted as cellulose chain-7 and chain-8 (see Figure 1A). The electrostatic energy associated with the crystallization process varies in the range of 50–150 kJ/mol, whereas vdW interaction energy becomes relevant in the case of the interaction with chain-7. The electrostatic interaction guides the interaction with chain-8. The same effect is observed for oligomer-4, which also interacts with two chains, i.e., chain-17 and chain-18, located at one of the hydrophilic crystal faces, i.e., $\left(\bar{1}10\right)$. These results highlight the interdependence between vdW and electrostatic energies during the recognition and binding processes. Stiffer oligomers recognize hydrophilic crystal faces better than hydrophobic ones, as they can form extended cellulose sheets rather than a new cellulose sheet.

Figure 6 shows the non-bonded energy profiles for semi-rigid 12 β-D-glucose monomers (i.e., 12-mers). We can observe a clear distinction in the energy landscape with respect to the previous oligomer (i.e., 6-mers). The additional chain flexibility facilitates the relaxation of the non-bonded energies, and in this case, the energy scales are more comparable with fluctuation in the range of 50–200 kJ/mol. In addition, here, two oligomers, namely 3 and 4, interact with the hydrophobic (100) and hydrophilic $\left(\bar{1}10\right)$ crystal layers, respectively. The inherent flexibility of the 12-mers allows the recognition of the hydrophobic layer, with an energetic cost in terms of vdW energy that is larger for the binding of hydrophobic crystal planes; in contrast, there is an energy scale quite similar to that of the stiffer 6-mers when it interacts with hydrophilic planes.

Figure 7 shows the analysis for the most flexible oligomer (i.e., 20 β-D-glucose monomers). According to the non-bonded energy, the energy scale for each non-bonded contribution is very similar to the semi-flexible 12-mers. Furthermore, we can account for a larger number of interactions with cellulose chains for this oligomer. For instance, oligomer-1 binds strongly to cellulose chain-1, chain-2, and chain-3, while oligomer-2 recognizes chain-12 and chain-13, and oligomer-4 is found to bind to chain-2 and chain-7. Here, we can also find interactions with hydrophobic (100) and hydrophilic planes (e.g., the (110) and $\left(1\bar{1}0\right)$ crystal layers) (see Figure 1A). To further verify the role of stiffness during the binding process between a crystallized protofibril and a cellulose I microfibril, we investigated their energy landscape (see Figure 8). In our study, this process is mediated by chain–chain interactions occurring at the interface of the two crystalline fibrils. We can see here that both non-bonded interactions, i.e., vdW and electrostatics, contribute to the stickiness of the protofibril–microfibril complex. This interaction is between chain-12 (in the microfibril) and chain-7 (in the protofibril), which could correspond to a sheet–sheet interaction via two (110) crystallographic planes.

### 2.3. Hydrogen Bond Analysis

Figure 9 (panels A and B) shows the MD simulation of the HB calculation during the crystallization process of the six β-D-glucose monomers onto crystalline cellulose I layers. In our case, we noticed the strong interaction with the crystal layers $\left(1\bar{1}0\right)$ and $\left(\bar{1}10\right).$ Both crystal planes are considered very hydrophilic. We found an average HB number ranging from four to seven HBs (see Table 1). For each case, i.e., the oligomer-1:chain-8 and oligomer-4:chain-18 complexes, we see a good correlation with an increase in the electrostatic energy (see Figure 5). The conformation of the bound oligomer is quite similar to a ribbon-like conformation, which is typical of cellulose chains in the microfibril. The 6-mers follow crystallization upon binding. Typical intersheet HBs, such as O6H⋯O2 (see Figure 9C), characterize the bound state. In several cases, 6-mers play the role of a donor molecule, with O2 and O6 as donor atoms. The O3 atom in the 6-mers is found to be an acceptor atom, and it does establish an O6H⋯O3 HB type.

Figure 10 (panels A and B) shows HB profiles for the semi-flexible oligomer (i.e., 12-mers). In this case, we can observe a larger number of HBs (~eight HBs) in comparison with the 6-mers case. A few of those HBs are formed between the beginning and ending residues in the oligomer (see Figure 10C) and are listed in Table 2. The oligomer follows a disorder-like conformation in the central part of the molecule (with no HB), whereas it shows crystallization at both ends. The latter is associated with the formation of HBs. As discussed before, a large contribution of electrostatic energy is required to keep the oligomer on the crystal plane $\left(\bar{1}10\right)$ and to establish intersheet HBs of the same type as O2H⋯O6 and O6H⋯O2. We also found some hydrophobic C_n_H⋯O_m_ HBs, which are weak and typically associated with the flexibility of the oligomer. In general, the oligomer plays a major role as a donor.

Figure 11 (panels A and B) describes the total number of HBs for the most flexible oligomer (i.e., 20-mers) in our study. In this case, the correct balance between the electrostatic and vdW energies leads to the formation of a large number of HBs, ranging from 9 to 20 (see Table 3). Figure 11C displays an MD snapshot that shows the interacting oligomer regions of both systems. The formation of HBs in the oligomer leads to local crystallization, with a chain conformation similar to that of the cellulose I chain. Due to the flexibility of the 20-mers, we can see a substantial number of HBs along the oligomer backbone that are not present in the other oligomer cases. The 20-mers becomes a donor for HB types O4H⋯O6, O3H⋯O6, O3H⋯O3, and O6H⋯O3. Furthermore, it is also engaged as a donor in establishing several C_n_H⋯O_m_ HBs. According to Table 3, the interaction with the (110) plane results in the formation of several HBs where the oligomer plays the role of acceptor, whereas interaction with the crystal plane (100) defines the donor character of the 20-mers.

Figure 12 reports on the protofibril–microfibril interaction that is related to the formation of a solid–solid interface in the MD simulation. This process may relate to the inherent stiffness of the protofibril and the long relaxation time required for structural changes at the interface of the complex. Here, we account for ~11 HBs (see Table 4). The protofibril and microfibril interact through their hydrophilic crystal planes (110). In this case, the protofibril plays the role of donor for the establishment of HBs. This result shows the effect of the stiffness on the HB chemistry. In the case of the oligomer, the interaction with hydrophilic crystal planes defines the acceptor character of the atoms in the oligomer, although this is not the same for already-crystallized systems, such as the protofibril. Due to the time scale of our simulations, we only describe the early step in the oligomer/protofibril crystallization on the cellulose I microfibril surface.

## 3. Materials and Method

### 3.1. Initial Structures

The cellulose chains (i.e., oligomers) and microfibrils are among the first two levels of the structural hierarchy of natural fibers. Cellulose microfibrils are frequently observed in bundles or fibers [26,27,28]. In fact, a cellulose microfibril is formed from the crystallization process of single oligomer chains. Thus, the modeling and characterization of oligomer–microfibril/protofibril–microfibril in MD led us to the choice of following the initial structures. The structure of the cellulose I microfibril, with a cross-section formed by 36 cellulose chains, was modeled by a cellulose-builder toolkit [29]. We built cellulose Iβ polymorph microfibrils with 40 β-D-glucose residues (or 20 cellobiose units) connected by β(1→4) glycosidic bonds that define the covalent bond between the oxygen atom and the C_1_ of one glucose ring and the C_4_ of the subsequent ring. The length of the protofibril was sufficient to show no deviation in the crystalline state during the MD simulation. To avoid the center of mass diffusion and rotation of the cellulose microfibril in the simulation box, we decided to apply forces that could mimic the position restraints of β-D-glucose monomers in residue 21, located in one of the cross-sections of the microfibril. We applied a force constant equal to 1000 kJ mol^−1^ nm^−1^ for each cartesian coordinate. The four β-D-glucose monomers located around the microfibrils were prepared by the Packmol package [30]. They were positioned along the main microfibril axis, which is parallel to the z-axis (see Figure 1B).

### 3.2. All-Atom MD Simulation

The initial structures of the native cellulose I microfibril (Iβ) were designed by the cellulose-builder toolkit [29]. In practice, we prepared the system with one cellulose microfibril center in the simulation box. The microfibril consisted of 36 cellulose chains, each one comprising 40 β-D-glucose units (i.e., 20 cellobiose units). The cellulose I microfibril and β-D-glucose oligomers were modeled by the CHARMM36c force field [31] in the GROMACS v2020.4 package [32]. The length of the oligomers varied in the range of 6, 12, and 20 β-D-glucose monomers; the monomers were parametrized in this force field. They were used to describe different regimes of flexibility. A triclinic box was used to represent the MD simulation box, and periodic boundary conditions (PBCs) were implemented for all cartesian coordinates. Solvation of the simulation box by TIP3P [33] water molecules with a buffer distance of 15 Å from the outer layer of the microfibril was considered to avoid any interaction between single oligomers and periodic images of the microfibril. A total of 12,673 water molecules were necessary for the solvation of the protofibril–microfibril systems. In the case of the oligomer–microfibril complexes, the numbers of water molecules were 128,631, 173,016, and 173,226, which correlated to 20 β-D-glucose monomers, 12 β-D-glucose monomers, and 6 β-D-glucose monomers.

The equilibration protocol started with over 1300 steps of energy minimization via a conjugate gradient method for the solvent, while the solute remained restrained. Then, an unrestrained MD simulation was conducted using a time step of 2 fs, a reference temperature of 300 K, and a pressure of 1 atm. To maintain the reference temperature and pressure, a v-rescale thermostat [34] and a Berendsen barostat [35] were employed for constant volume (NVT) and constant pressure (NPT) ensembles. Long-range electrostatic interactions were computed using the PME methodology. Finally, an MD production run in the NVT ensemble, with about 60 ns for each system, was conducted, and two independent trajectories were analyzed. Figure 1 (panels B and C) shows the initial and final positions of the oligomers and the cellulose I microfibril.

### 3.3. Energetic Analysis of the Aggregation Pathway

The non-bonded energies, such as the van der Waals and electrostatic interactions, were calculated in the equilibrated MD trajectories, and they allowed the characterization of the crystallization process for an oligomer on a cellulose microfibril surface. We decided on a cutoff radius of 1.0 nm, and the calculation of the long-range interaction for the electrostatic energy calculation employed the PME method.

### 3.4. Fluctuation Analysis of the Bound Complex by Root Mean Square Fluctuation

Monomer fluctuations for each oligomer were computed through the root mean square fluctuation (RMSF) profile using the GROMACS gmx tool analysis. We performed this analysis for 6, 12, and 20 β-D-glucose monomers. The RMSF can reveal which areas of a system are the most mobile; for instance, an area of the structure with high RMSF values frequently diverges from the average structure, indicating high mobility. In the case of proteins, RMSF is computed either for heavy atoms in the backbone or for the C_ɑ_ atoms, as they provide good information on the conformational changes in the protein. In case of oligomer fluctuations, we employed the position of heavy atoms denoted as C1, C2, C3, C4, C5, C6, O2, O4, O5, and O6.

### 3.5. Hydrogen Bond Calculations

A hydrogen bond (HB) is formed when the donor (D) and the acceptor (A) atoms are separated by a distance less than 3.5 Å, the distance between the hydrogen (H) and A atoms is simultaneously less than 2.7 Å, and, finally, the angle θ formed between H, D, and A is less than 30 degrees. The HB type was defined based on the hydrophobic and hydrophilic character of the bond, which was based on the interaction with hydrophobic, e.g., (100) and $\left(\bar{2}00\right)$, and hydrophilic, e.g., $\left(\bar{1}10\right)$, (110), $\left(1\bar{1}0\right)$ and $\left(\bar{1}\bar{1}0\right)$, cellulose layers (see Figure 1A).

## 4. Conclusions

The self-assembly process of β-D-glucose oligomers is still poorly understood due to the time and length scales associated with it. In this study, we described the interaction of oligomers/protofibril with different surfaces of the cellulose I microfibril. Oligomer crystallization was found to be facilitated by chain stiffness. In this regard, the stiffer the oligomer was, the stronger the intersheet HBs of the type O2H⋯O6 and O6H⋯O2 that were formed at the interface of the complex. The analysis by MD simulations, in terms of the length and flexibility of the oligomers, showed consistency for the 6-mers. This process was facilitated by the fast structural relaxation of the smaller oligomers and the formation of HBs in the hydrophilic crystal planes. The analysis of non-bonded energies showed the interplay between vdW and electrostatic interaction during crystallization. The energy scale was dominated by electrostatic energy in the presence of strong HBs (i.e., O2H⋯O6 and O6H⋯O2). Other types of HB were established in hydrophilic interfaces, such as $\left(\bar{1}10\right)$, $\left(\bar{1}\bar{1}0\right)$, $\left(1\bar{1}0\right)$, and (110); generally, they were accessible due to the flexibility of the oligomer (i.e., 12-mers and 20-mers). The recognition process of an assembled protofibril with a cellulose I microfibril in solution was mediated by the vdW and electrostatic interactions at first; then, such cases should follow the local rearrangement of the protofibril chains at the interface dominated by vdW energy. In our case, the formation of few HBs between two hydrophilic crystal faces (e.g., (110)) resulted in the donor character of the protofibril. Our findings, in terms of the self-assembly of a cellulose microfibril from a molecular point of view, will assist the design of novel cellulose-based materials (e.g., composites) with exceptional mechanical and structural properties and will also assist with the optimization of cellulose degradation or modification under enzymatic treatments (e.g., enzymatic cocktails and cellulosome principles) that can support biotechnology research in biofuels, renewable and biodegradable materials, etc. In addition, our results offer new molecular insight into the assembly process that could be incorporated into coarse-grained MD methodologies, such as our recent MARTINI 3 cellulose force field [36]; in this way, we could aim to extend the time scales to several microseconds and also describe very large-scale systems in complexes with other biomolecules in MD simulations [37].

## Figures and Tables

**Figure 1 ijms-23-08505-f001:**
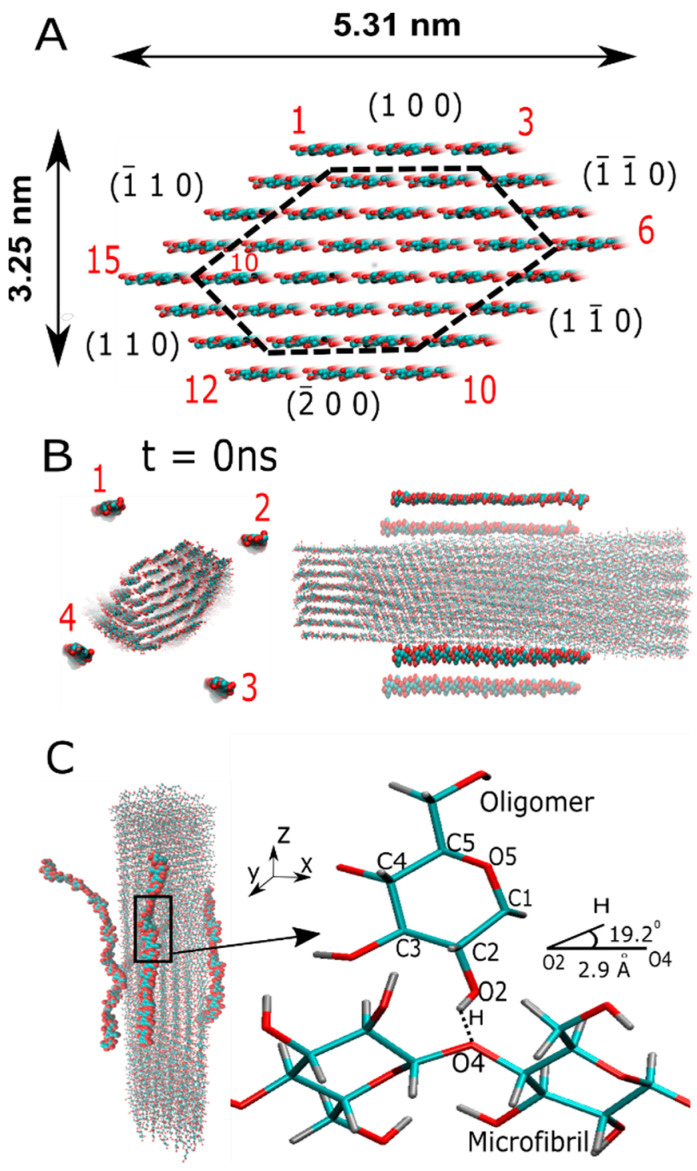
The molecular structure of the cellulose Iβ microfibril and β-D-glucose oligomers are depicted without the solvent. The hydrogen atoms are not represented for a better visualization. Panel (**A**) shows the cellulose I microfibril and typical length dimensions. Crystal planes are indicated by the Miller nomenclature (h,k,l) [25], and these labels are located next to each crystal plane, represented in red and in a clockwise manner. The protofibril is highlighted within the dashed lines, and the chain label of the outer layer runs as before. Panel (**B**) shows the initial position (at t = 0 ns) of the β-D-glucose oligomers in the XY plane and along the main microfibril axis. Panel (**C**) shows an MD snapshot of the oligomer–microfibril complex. The oligomers are crystallized along the main microfibril axis (i.e., the *z*-axis). The right side of this panel depicts a zoom in the highlighted region in the presence of hydrogen bonds. HB parameters are shown.

**Figure 2 ijms-23-08505-f002:**
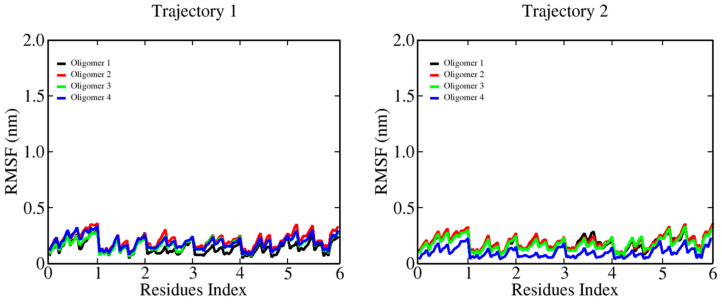
Root mean square fluctuations (RMSFs) of the β-D-glucose oligomers (6-mers) in complex with a cellulose I microfibril. For each β-D-glucose oligomer, the RMSF profile is shown as a colored solid line (see the inset).

**Figure 3 ijms-23-08505-f003:**
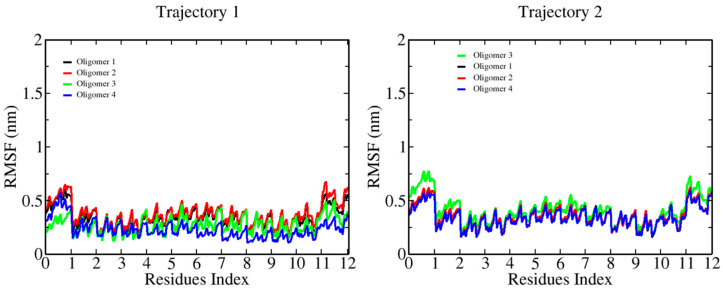
Root mean square fluctuations (RMSFs) of the β-D-glucose oligomers (12-mers) in complex with a cellulose I microfibril. RMSF profiles are colored as in Figure 2.

**Figure 4 ijms-23-08505-f004:**
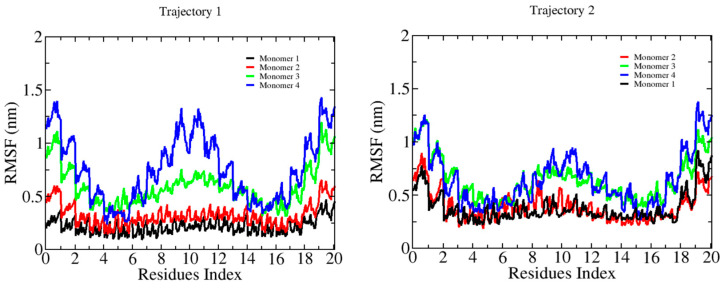
Root mean square fluctuations (RMSFs) of the β-D-glucose oligomers (20-mers) in complex with a cellulose I microfibril. RMSF profiles are colored as in Figure 2.

**Figure 5 ijms-23-08505-f005:**
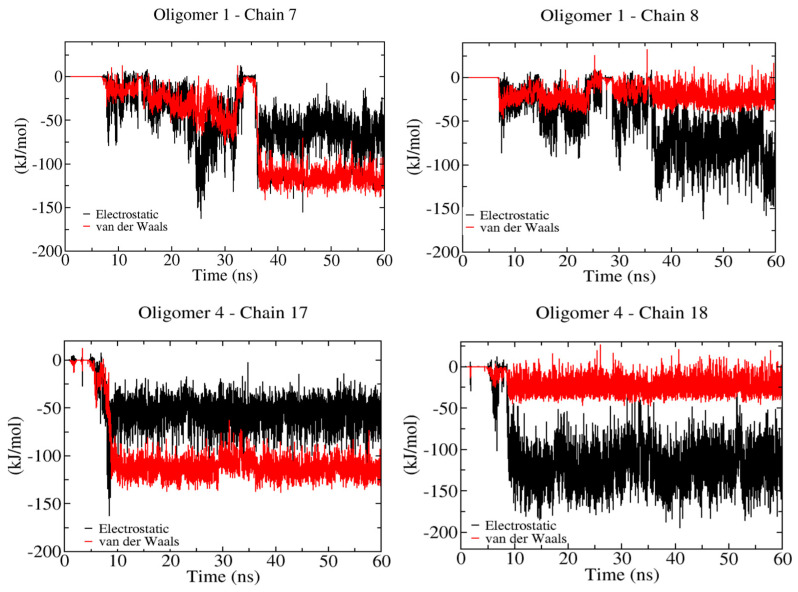
Non–bonded contributions during the binding of six β-D-glucose monomers onto a cellulose I microfibril. Solid red and black lines correspond to vdW and electrostatic energies, respectively.

**Figure 6 ijms-23-08505-f006:**
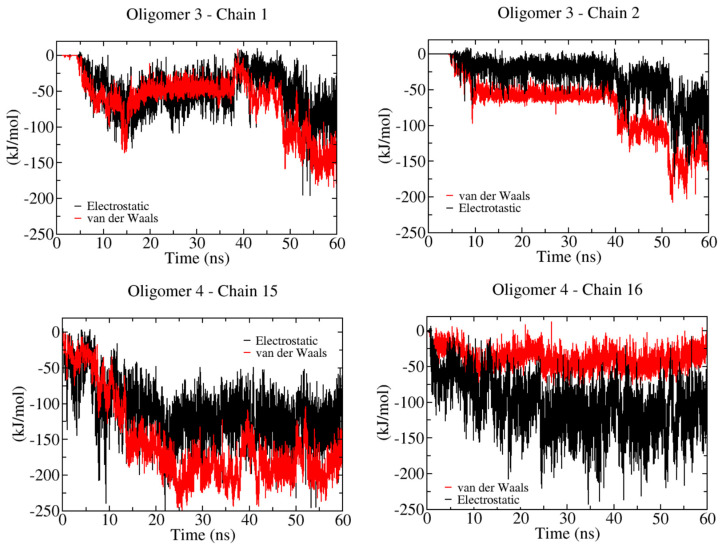
Non−bonded energy contributions during the binding of 12 β-D-glucose monomers onto a cellulose I microfibril. Solid red and black lines correspond to vdW and electrostatic energies, respectively.

**Figure 7 ijms-23-08505-f007:**
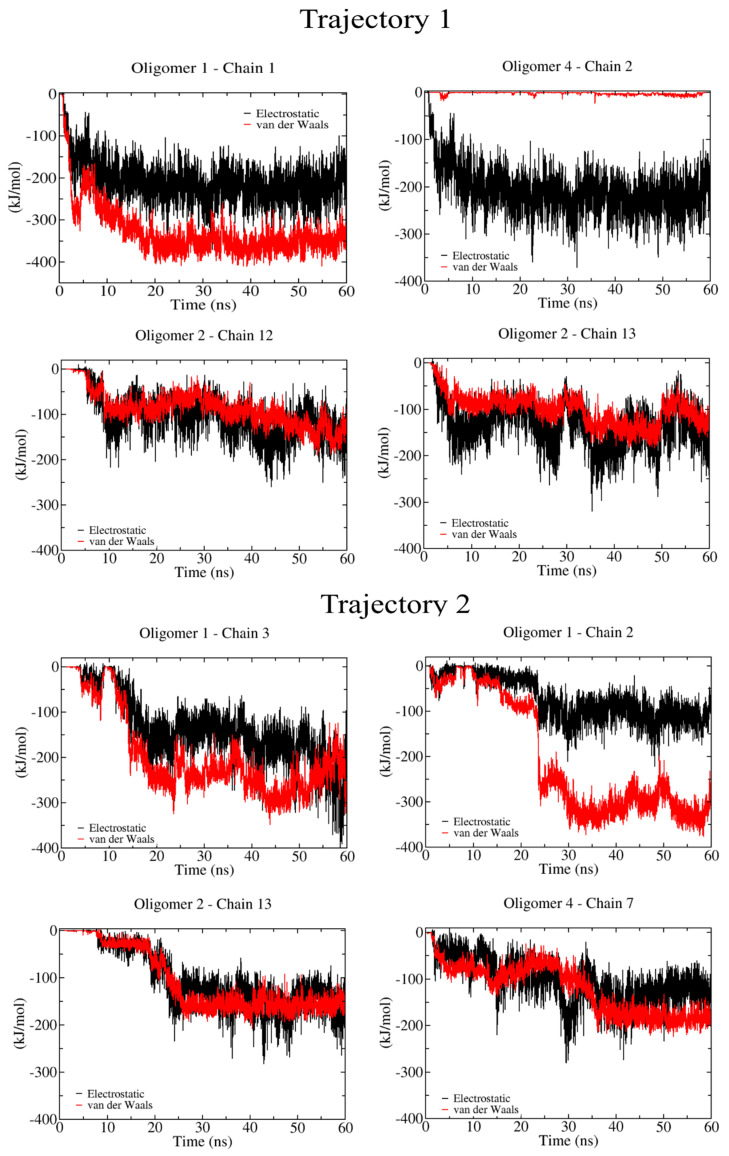
Non−bonded energy contributions during the binding of 20 β-D-glucose monomers onto a cellulose I microfibril. Solid red and black lines correspond to vdW and electrostatic energies, respectively.

**Figure 8 ijms-23-08505-f008:**
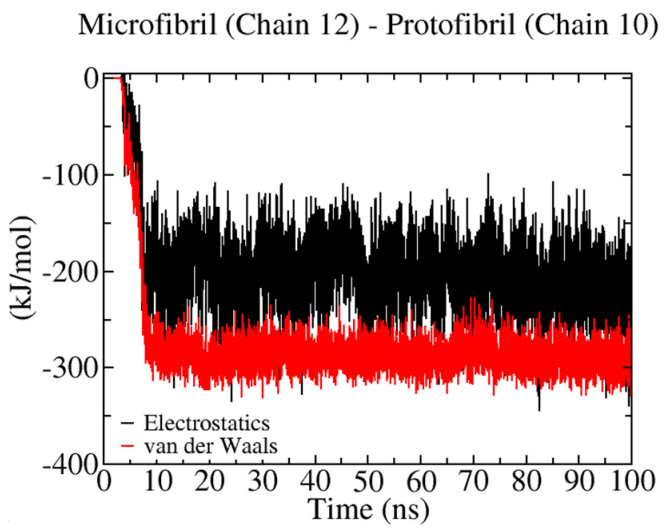
Non−bonded energy contributions during the binding of a protofibril and cellulose I microfibril. Solid red and black lines correspond to vdW and electrostatic energies, respectively.

**Figure 9 ijms-23-08505-f009:**
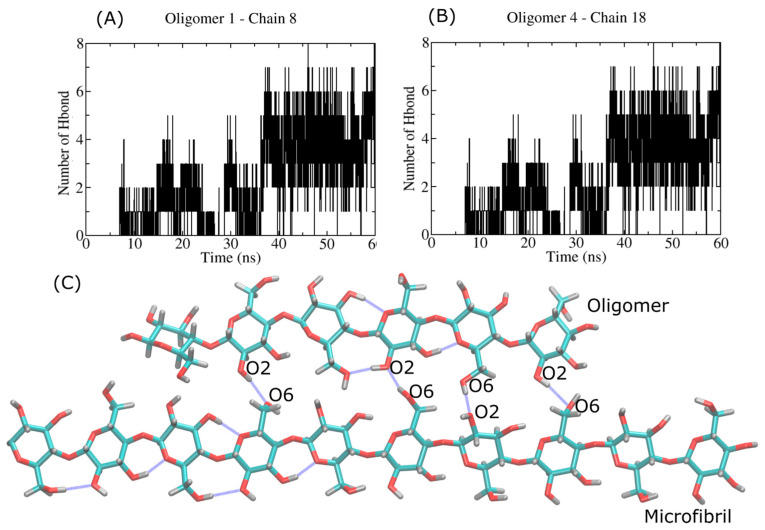
Hydrogen bond profile of the interaction between the smallest oligomer (6 β-D-glucose monomers) and the cellulose I microfibril. Panel (**A**,**B**) shows the HB profiles for the oligomer-1:chain-8 and oligomer-4:chain-18 complexes. Panel (**C**) shows the MD snapshot of the bound complex, with HBs highlighted by blue lines.

**Figure 10 ijms-23-08505-f010:**
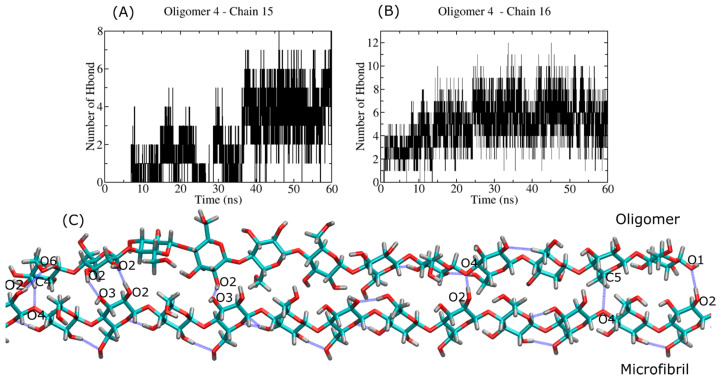
Hydrogen bond profile of the interaction between the semi-flexible oligomer (12 β-D-glucose monomers) and the cellulose I microfibril. Panel (**A**,**B**) shows the HB profiles for the oligomer-4:chain-15 and oligomer-4:chain-16 complexes. Panel (**C**) shows the MD snapshot of the bound complex, with HBs highlighted by blue lines.

**Figure 11 ijms-23-08505-f011:**
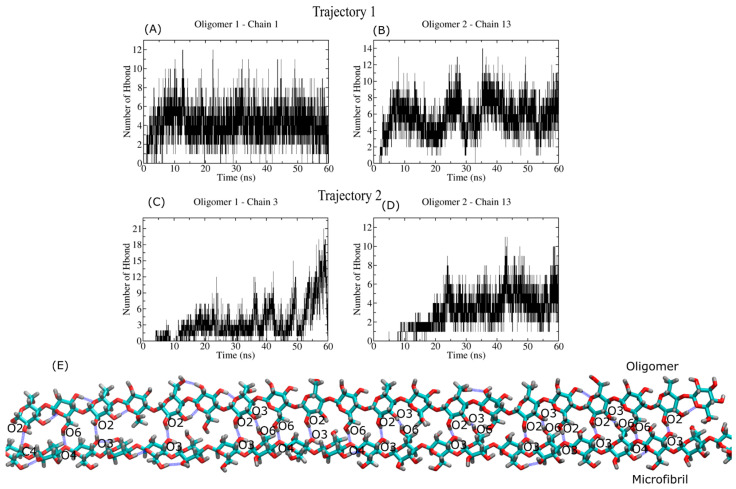
Hydrogen bond profile of the interaction between the very flexible oligomer (20 β-D-glucose monomers) and the cellulose I microfibril. Panel (**A**–**D**) shows the HB profiles for different oligomer:chain complexes. Panel (**E**) shows the MD snapshot of a bound complex, with HBs highlighted with blue lines.

**Figure 12 ijms-23-08505-f012:**
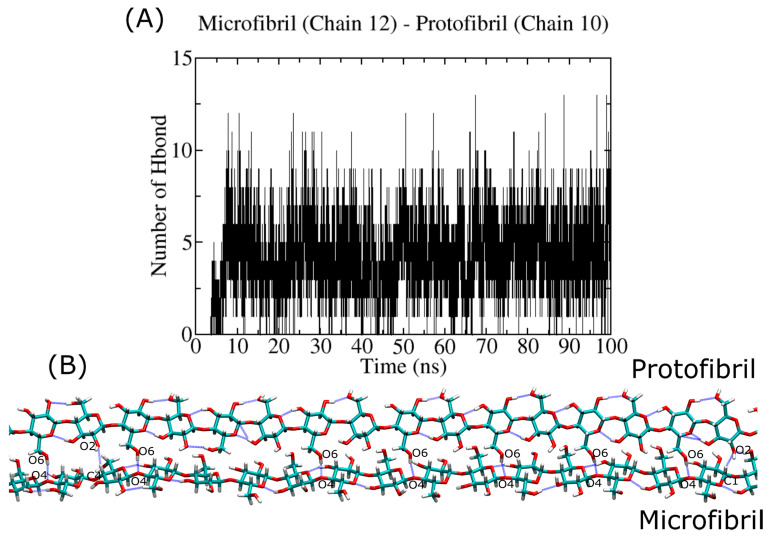
Hydrogen bond profile of the interaction between the cellulose protofibril (composed of 18 chains) and the cellulose I microfibril. Panel (**A**) shows the HB profiles for the protofibril:microfibril complex. Panel (**B**) shows an MD snapshot of the complex, with HBs highlighted with blue lines.

**Table 1 ijms-23-08505-t001:** Hydrogen bond types between the six β-D-glucose monomers and the cellulose I microfibril. In square brackets are highlighted the positions of the oligomer and microfibril residues, denoted by the letters I and F, respectively, that are involved in HBs.

Oligomer	Chain in Microfibril	HB Type (Donor-Acceptor)
**Trajectory 1**		
1	8	O6H⋯O2 [F5–I3] O2H⋯O6 [I1–F3] O6H⋯O2 [I2–F4] O2H⋯O6 [I5–F7]
**Trajectory 2**		
4	18	O3-H⋯O6 [F1–I6] O6-H⋯O3[F4–I3] O2-H⋯O6[F5–I2] O6-H⋯O3[F6–I1] O2-H⋯O6 [I5–F2] O2-H⋯C6 [I5–F2] O6-H⋯O3 [I2–F5]

**Table 2 ijms-23-08505-t002:** Hydrogen bond types between the 12 β-D-glucose monomers and the cellulose I microfibril. In square brackets are highlighted the positions of the oligomer and microfibril residues, denoted by the letters I and F, respectively, that are involved in HBs.

Oligomer	Chain in Microfibril	HB Type (Donor-Acceptor)	
4	15	O2H⋯O6 [F15–I1] O2H⋯O2 [F17–I2] O2H⋯O4 [F23–I9] O2H⋯O1 [F27–I12] O3H⋯O3 [I2–F17] O2H⋯O3 [I4–F19]	C5H⋯O4 [I11–F26] C4H⋯O4 [I1–F16]
4	16	O2H⋯O3 [F15–I1] O2H⋯O6 [F19–I5] O2H⋯O6 [F25–I11] O2H⋯O6 [I1–F16] O2H⋯O6 [I4–F18] O3H⋯O6 [I6–F20] O2H⋯O6 [I10–F24] O2H⋯O6 [I12–F26]	

**Table 3 ijms-23-08505-t003:** Hydrogen bond types between the 20 β-D-glucose monomers and the cellulose I microfibril. In square brackets are highlighted the positions of the oligomer and microfibril residues, denoted by the letters I and F, respectively, that are involved in HBs.

Oligomer	Chain in Microfibril	Type (Donor-Acceptor)	
**Trajectory 1**			
1	1	O2H⋯O3 [F13–I2] O6H⋯O2 [F14–I2] O6H⋯O2 [F22–I10] O2H⋯O6 [F25–I14] O6H⋯O2 [F32–I20] O4H⋯O6 [I1–F12] O3H⋯O6 [I10–F21] O2H⋯O4 [I12–F24] O6H⋯O6 [I16–F28]	C2H⋯O3 [I5–F17] C1H⋯O6 [I6–F18] C5H⋯O6 [I6–F18] C2H⋯O4 [F19–I8] C2H⋯O4 [I14–F26]
2	13	O3H⋯O3 [I8–F19] O6H⋯O2 [I14–F25] O6H⋯O3 [I16–F27] O6H⋯O3 [I20–F31] O3H⋯O6 [F17–I6] O2H⋯O2 [F21–I10] O2H⋯O2 [F23–I12] O6H⋯O6 [F24–I13] O2H⋯O3 [F25–I15] O6H⋯O4 [F28–I18]	
**Trajectory 2**			
1	3	O2H⋯O3 [I2–F17] O6H⋯O4 [I3–F18] O2H⋯O3 [I4–F19] O6H⋯O4 [I5–F20] O2H⋯O3 [I6–F21] O2H⋯O3 [I8–F23] O2H⋯O3 [I10–F25] O6H⋯O4 [I11–F26] O2H⋯O3 [I12–F27] O6H⋯O4 [I13–F28] O2H⋯O3 [I14–F29] O2H⋯O3 [I16–F31] O6H⋯O4 [I19–F34] O6H⋯O3 [F18–I4] O6H⋯O3 [F20–I6] O6H⋯O3 [F22–I8] O6H⋯O3 [F24–I10] O6H⋯O3 [F28–I14] O3H⋯O2 [F33–I18]	C4H⋯O2 [F35–I20]
2	13	O6H⋯O3 [I9–F19] O6H⋯O2 [I18–F27] O6H⋯O2 [F18–I8] O6H⋯O4 [F20–I11]	

**Table 4 ijms-23-08505-t004:** Hydrogen bond types between a protofibril and cellulose I microfibril. In square brackets are highlighted the positions of the protofibril residues and microfibril residues, denoted by the letters P and F, respectively.

Chain in Protofibril	Chain in Microfibril	HB Type (Donor-Acceptor)	
7	12	O6H⋯O4 [P2–F13] O6H⋯O4 [P4–F15] O6H⋯O4 [P8–F19] O6H⋯O4 [P10–F21] O6H⋯O4 [P12–F23] O6H⋯O4 [P14–F25] O6H⋯O4 [P16–F27] O6H⋯O4 [P18–F29] O6H⋯O2[F28–P17]	C4H⋯O2 [F14–P3] C1H⋯O2 [F27–P17]

## Data Availability

Not applicable.

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
