# Peer review of "Molecular Insight into the Self-Assembly Process of Cellulose Iβ Microfibril"

_ijms, 2022, doi:10.3390/ijms23158505_

Round 1

Reviewer 1 Report

In this work molecular insight into the self-assembly process of cellulose Iβ microfibril is described. The self-assembly process of β-D-glucose oligomers on the surface of cellulose Iβ microfibril involves crystallization. The characterization of this process involves the structural relaxation of the oligomer, recognition of the cellulose I microfibril and the formation of several hydrogen bonds. It was shown that the oligomer crystallization is facilitated by the chain stiffness. The work is of interest because offers new molecular insight into the assembly process which could be incorporated in coarse-grained Molecular dynamics methodologies. The article looks like a short communication and may be published after minor revision.

Notes:

1. The meaning of “NPT and NVT” abbreviations should be added in the text. (line 166)

2. How a quantification of the most relevant HB types between the oligomer and crystalline microfibril does influence on the oligomer crystallization?

3. In what areas of industry will new materials based on cellulose be in demand based on the findings of this work? It should be mentioned more detailed.

Reviewer 2 Report

The manuscript “Molecular insight into the self-assembly process of cellulose Iβ microfibril” by Tran Thu Minh Thu et al. presents interesting theoretical work on cellulose advanced structure formation. Using molecular dynamics simulations, the authors investigate the factors affecting association and crystallization of oligomers.

The project is well described in the Abstract, giving good invitation to the full paper. The experiments are presented with figures and descriptions, and sufficient summary. The idea of the study is based on the available data and its future application is explained. The benefits from the study for further research and practice may be discussed more in the conclusions.

The idea of the paper and specific results get more clear after reading the whole text. 

There are questions and issues that should be addressed before further proceedings with this paper.

1. Please remind in more places in the text that this is a MD study, as a sentence (line 27-28) “In addition, we study the interaction between a protofibril and the cellulose I microfibril in solution.” is an example that some fragments are suggesting a study on cellulose by X-ray or in actual solution.

2. Line 71-72: “molecular networks of hydrogen bonds (HB) between β-D-glucose chains established by the electrostatic interaction of the negatively charged  acceptor and positively charged donor [16]”. Is there a formal charge in glucose, or rather electron density shift due to polarization of the bonds?

3. Line 87: “we shed light onto the chemistry of oligomer-microfibril assembly”. Is this phenomenon actually a chemical process, physical or physicochemical one?

4. Line 112: How was the length of protofibril selected  (protofibril with a length of 20 β-D-glucose monomers)? The oligomers were 6, 12 and 20 unit chains – please indicate the factors affecting this choice.

5. Please consider a more precise description of restrains in oligomer and microfibril in Introduction,: Line 122: “The oligomers are not restrained during the MD simulation” as then line 132 created a confusion: “we decided to restrain the position of residues 21 located”

6. When referring to Figure 1 in the text, please indicate the panel.

7. Line 185: To avoid confusion, please list the heavy atoms.

8. Line 189: the angle of HB: finally the angle θ formed between D, H and A is less than 135 degrees. It would be a good idea to provide one more figure, showing small fragment of glucose oligomer with atoms numbered, hydrophilic-hydrophobic faces indicated  and an example of HB. For most readers not associated with this topic (or biomolecular structural investigations) glucopyranose is probably flat and just hydrophilic....

 9. The structures of glucose rings in Figures 9 etc. are rather small and finding the indicated features is difficult. The markings in Figure 10C are not easy to associate with specific atoms. As the analyzed oligomers get bigger, it is more difficult to fit them in the figure, but the current version is not a very helpful one.

10. Table 1. Please verify the Trajectory 2  as carbon is rarely the acceptor of HB “O2-H⋯C6 [I5–F2] “. It is also the only Table with different presentation of HB donors “O6H⋯O2” and “O6-H⋯O3”.

11. Line 355: Please add what is interacting with microfibril: “In this study we describe the interaction with different surfaces in the cellulose I microfibril.”

Minor issues:

1. The problem with general style affecting clarity:

Line 33: In some places the sentences are rather long and combine concepts in a rather risky way. The sentence describing cellulose from glucose unit to multifiber product is an example “which is the result of the self-assembly process of linear chains of β-D-glucose connecting carbons C1 and C4  atoms via the glycosidic bond centered at the connecting oxygen atom...”. As it is mentioned later, cellulose biosynthesis is enzymatic process at the level of glucose unit connecting, and the self-assembly plays a role in higher-order structures.(“where the conversion of a single β-D-glucose monomer to a cellulose polymer requires several enzymatic steps”).

Another convoluted sentence: “starts by phosphorylation of a single monomer to the uridine diphosphate glucose (UDP-glucose) by UDP-glucose pyrophosphorylase.” Phosphorylation is commonly recognized as attachment of phosphate to other molecule (ester formation). Please consider rephrasing this sentence. Is UDP-glucose a substrate or a product?

Another example, line 100: “Once the oligomer is weakly bounded on the microfibril surface, then follow conformational changes of the oligomer along the main microfibril axis and final dissipation of the intrinsic flexibility of the oligomer on the a crystalline surface will allow its crystallization.”. It is a long and complex sentence with advanced grammar structure, which does not help in reading and understanding it. Please consider rephrasing (separation, passive voice?)

Line 104: “forces and electrostatic contributions, including single chain fluctuations in solution and once is adsorbed in the microfibril” – what is absorbed?

Line 154: “The length of oligomers varies in the range of 6, 12 and 20 β-D-glucose monomers, denoted as BLGC residue in this force field, were used to describe different regimes of flexibility.”. Is there something is missing in this sentence? Please explain BLGC.

2. Please decide on one form of “Van der Waals (VdW) or van der Waals (vdW)” in the text.

3. Line 92: hydroxymethyl (–CH2OH) – this is probably the only one place in the text when the “2” locant should be in subscript.

4. Line 30: “the protofibril becomes a donor of those HBs.” – here HB origin is associated with hydrogen atom, however, HB donor is also a particular functional group. For a moment I was considering “a donor in HBs”, but this statement in Abstract is general, whereas in the main text authors describe HB in detail.

5. Please verify the grammar tenses used in the text, as present is used both for general truths and experimental setups (for example part 2.3).

6. Figure captions: The idea with square brackets is not optimal  “In  [.] are highlighted the position”

Reviewer 3 Report

This paper presented an MD investigation of the self-assembly process of cellulose Iβ 2 microfibril.  The authors first reviewed the state of the art. Then, the MD model was introduced. The authors reported adequate results in this paper. However, clarifications are suggested by the reviewer before publishing this paper. Please find the comments and questions below.

1. The novelty of this paper should be clarified in the introduction. Why the selected topic is important and should be studied? What is the new knowledge in this reported work?

2. Please explain why the initial structures were created in the reported way (section 2.2). Why a certain number of water molecules was used? 

3. Please better explain the self-assembly process simulated using MD modeling in the results section. 
